# Reproducibility study of the Value Equivalence principle for Model Based Reinforcement Learning

## Reproducibility Summary

**Scope of Reproducibility**

Grimm et al. [2020] introduces and studies the concept of equivalence for Reinforcement Learning models with respect to a set of policies and value functions. It further shows that this principle can be leveraged to find models constrained by representational capacity, which are better than their maximum likelihood counterparts.

**Methodology**

The code for this project is closed sourced. Therefore, we re-implemented the three sets of experiments (including the baseline) and the authors' custom environments. All experiments were performed using Google Colab and required a total of 160 hours of Google Colab GPU.

**Results**

Since all the results in the original paper are presented in graphical form, we cannot provide precise numbers. For experiments with $\mathrm{span}(\mathcal{V}) \approx \tilde{\mathcal{V}}$, our results match the reported results. For experiments with $\mathrm{span}(\mathcal{V}) \approx \tilde{\mathcal{V}}$ and linear function approximation, our results for both the baseline and the author's method diverge from the reported graphs. For experiments with $\mathrm{span}(\mathcal{V}) \approx \tilde{\mathcal{V}}$ and neural networks, our results follow the reported trend, but not always with the same values.

**What was easy**

Even though we had to re-implement everything from scratch, the general pipeline for all experiments was straightforward and well described in the original paper. The environments used for all three experiments were reasonably straightforward.

**What was difficult**

Grimm et al. [2020] is a combination of theory and experiments. It is crucial to understand the theorems presented in the paper, requiring a solid knowledge of linear algebra. For experiments with linear function approximation, feature selection was made using k-means which depends heavily on centroids' initialisations. It takes a significant amount of time to repeatedly apply k-means on a large set of data to find the best fit (more than 10 hours for 10,000 initialisations on a dataset of 1,000,000). For these experiments, we also found that the given learning rate did not learn a good model (see figure 1).

**Communication with original authors**

We contacted the author for multiple queries related to custom environments, hyper-parameters, feature selections and other minute experimental details via email. The author replied to all of them thoroughly and in a reasonable time.

# 1 Introduction

[Grimm et al. [2020]] introduces the principle of Value Equivalent models with respect to a set of policies and value functions. Consider, $\Pi$ to be a set of policies and $\mathcal{V}$ to be a set of functions from $S \to R$. Any two models $m$ and $\tilde{m}$ are value equivalent with respect to $\Pi$ and $\mathcal{V}$ if and only if, for all $\pi \in \Pi$ and all $\nu \in \mathcal{V}$, the following is true.

$$\mathcal{T}_\pi \nu = \tilde{\mathcal{T}}_\pi \nu$$

$T_\pi$ is the bellman operator of the MDP with respect to policy $\pi$.

The space of models which are value equivalent to the true model of the MDP, with respect to $\Pi$ and $\mathcal{V}$, is called $\mathcal{M}(\Pi, \mathcal{V})$. The main argument of the paper is that we can aptly choose $\mathcal{V}$ and $\Pi$ and exploit the Value Equivalence Principle to find models in $\mathcal{M}(\Pi, \mathcal{V})$ which are simpler and save computation and memory.

# 2 Scope of reproducibility

[Grimm et al. [2020]] shows that it is possible to exploit some structure and define the sets $\mathcal{V}$ and $\Pi$ for training models using the Value Equivalence principle. And that these models perform better than Maximum Likelihood models under rank constraints.

**Exploiting structure in the problem:** The set of true value functions of an MDP forms a polytype $\ddot{\mathcal{V}}$ ([Dadashi et al. [2019]]). To span as much of the polytype as possible, we define $\mathcal{V}$ to contain true value functions $\nu_\pi$ associated with random deterministic policies $\pi$.

**Exploiting structure in the solution:** In most applications of model based RL, value functions or policies are represented using function approximation. Suppose the agent can only represent policies $\pi \in \tilde{\Pi}$ and value functions $\nu \in \tilde{\mathcal{V}}$, then any model belonging to the set $\mathcal{M}(\tilde{\Pi}, \tilde{\mathcal{V}})$ is as good as the true model of the MDP.

We aim to verify these claims by using rank constrained models to compare the performance of Value Equivalence and Maximum Likelihood Estimation.

# 3 Methodology

We re-implement the training, evaluation, and custom environments for all the authors' experiments using the experimental pipeline and environment description provided in the appendix of [Grimm et al. [2020]].

## 3.1 Experimental setup and code

The total reproducibility can be divided into three type of experiments:

- $\mathrm{span}(\mathcal{V}) \approx \ddot{\mathcal{V}}$ and finite state space (4.1.1).
- $\mathrm{span}(\mathcal{V}) \approx \tilde{\mathcal{V}}$ and finite state space, using linear function approximation (4.1.2).
- $\mathrm{span}(\mathcal{V}) \approx \tilde{\mathcal{V}}$ and infinite state space, using neural networks (4.1.3).

We used the Catch environment for the first two experiments and the CartPole environment for the last one. The authors have used a custom Four-Rooms environment as well, which has been uploaded to our Github repository and can be easily used by future practitioners. Our entire code is available at [https://github.com/RajGhugare19/VE-principle-for-model-based-RL](https://github.com/RajGhugare19/VE-principle-for-model-based-RL)

## 3.2 Hyperparameters

The authors have provided all the hyper-parameters in the appendix. We found that for experiments with $\mathrm{span}(\mathcal{V}) \approx \tilde{\mathcal{V}}$ and linear function approximation, the specified learning rate ($5 \cdot 10^{-5}$) did not work. After doing a learning rate search, we found that $5 \cdot 10^{-2}$ works best for these experiments (see figure 1).

## 3.3 Computational requirements

We carried out all the experiments on the free GPUs from Google Colab. The GPU memory requirements and the total GPU hours required by every experiment are provided below.

Figure 1:

learning_rate search

Table 1: Compute Requirements per experiment

| Experiment | CUDA memory (MB) | GPU training time (hrs) |
|---|---|---|
| $\mathrm{span}(\mathcal{V}) \approx \tilde{\mathcal{V}}$ and infinite state space | 1600 | 8 |
| $\mathrm{span}(\mathcal{V}) \approx \tilde{\mathcal{V}}$ and finite state space | 600 | 0.25 |
| $\mathrm{span}(\mathcal{V}) \approx \ddot{\mathcal{V}}$ and finite state space | 560 | 0.25 |

## 4 Results

In experiments with $\mathrm{span}(\mathcal{V}) \approx \ddot{\mathcal{V}}$ and $\mathrm{span}(\mathcal{V}) \approx \tilde{\mathcal{V}}$ with neural networks, we found that the Value Equivalence models consistently performed better than the baseline, under rank constraints. For experiments with $\mathrm{span}(\mathcal{V}) \approx \tilde{\mathcal{V}}$ with linear function approximation we couldn't reproduce the results for both Value Equivalence training as well as the baseline. Although using a different evaluation technique for linear value function approximation, we show new results that convey the superiority of Value Equivalent models under constraints.

### 4.1 Results reproducing original paper

### 4.1.1 Result 1

The results from table 2 support the claim that Value Equivalence training converges towards a value equivalent model in a rank constrained space, i.e. a space with lesser span, when enforcing value equivalence with the value function polytype ($\mathrm{span}(\mathcal{V}) \approx \ddot{\mathcal{V}}$).

Table 2: $\mathrm{span}(\mathcal{V}) \approx \ddot{\mathcal{V}}$ and Catch environment

Table 3: rank 40

| size $\mathcal{V}$ | Our Results | | Reported Results | |
|---|---|---|---|---|
| | VE | MLE | VE | MLE |
| 40 | 9.898 | 4.929 | 9.8 | 4.8 |
| 35 | 9.879 | 4.929 | 9.8 | 4.8 |
| 30 | 9.883 | 4.929 | 9.8 | 4.8 |
| 25 | 9.894 | 4.929 | 9.8 | 4.8 |
| 20 | 9.817 | 4.929 | 9.8 | 4.8 |
| 15 | 8.876 | 4.929 | 9.8 | 4.8 |
| 10 | 8.527 | 4.929 | 8.9 | 4.8 |
| 5 | 7.720 | 4.929 | 7.9 | 4.8 |
| 2 | 6.926 | 4.929 | 6.3 | 4.8 |

Table 4: size $\mathcal{V}$ (# of policies) = 10

| rank | Our Results | | Reported Results | |
|---|---|---|---|---|
| | VE | MLE | VE | MLE |
| 20 | 9.479 | 4.303 | 8.5 | 5.1 |
| 30 | 9.883 | 5.975 | 9.4 | 4.3 |
| 40 | 8.527 | 4.929 | 9.6 | 6.2 |
| 50 | 8.914 | 8.002 | 9.8 | 9.1 |
| 100 | 9.049 | 9.887 | 9.8 | 9.6 |
| 150 | 9.089 | 9.910 | 9.9 | 9.9 |
| 200 | 9.895 | 9.910 | 9.9 | 9.9 |
| 250 | 8.053 | 9.910 | 9.9 | 9.9 |

The results are average state values corresponding to the policies formed using respective models.

 ### 4.1.2   Result 2

We followed the given pipeline and performed several experiments, the code for which can be found in our repository. However, the values corresponding to the resulting policies for both VE and MLE models were highly stochastic and did not reflect the authors' results.

To further evaluate the experiments with $\text{span}(\mathcal{V}) \approx \tilde{\mathcal{V}}$ and linear function approximation we use Double DQN where the action value function is a linear function approximator (see table 5).

The results clearly show that the Value Equivalence Principle can exploit the fact that the agent can only represent linear value functions. They also support the claim that under rank constraints, agents trained using the VE principle perform substantially better.

Table 5: Additional results for $\text{span}(\mathcal{V}) \approx \tilde{\mathcal{V}}$ and Catch environment

Table 6: size $\mathcal{V}$ (width) = 50

| rank | VE | MLE |
|------|------|------|
| 20 | 6.34 | 3.96 |
| 30 | 9.13 | 6.06 |
| 40 | 10.0 | 7.74 |
| 50 | 9.2 | 7.85 |
| 100 | 9.93 | 5.93 |
| 150 | 10.0 | 9.45 |
| 200 | 9.15 | 6.39 |
| 250 | 9.54 | 10.0 |

Table 7: rank 40

| size $\mathcal{V}$ | VE | MLE |
|------|------|------|
| 20 | 4.89 | 4.69 |
| 30 | 10.0 | 8.04 |
| 40 | 8.19 | 7.16 |
| 50 | 10.0 | 7.61 |
| 100 | 8.89 | 8.93 |
| 150 | 9.07 | 8.24 |
| 200 | 9.27 | 9.53 |
| 250 | 7.83 | 8.97 |

### 4.1.3   Result 3

The results from table 8 indicate that even when the set $\tilde{\mathcal{V}}$ consists of neural networks, the Value Equivalent models with rank constraints tend to perform better than the baseline. However, as we reduce the constraints, the performance of the VE models seems to deteriorate.

Table 8: $\text{span}(\mathcal{V}) \approx \tilde{\mathcal{V}}$ and CartPole environment

Table 9: rank 10

| size $\tilde{\mathcal{V}}$ | Our Results | | Reported Results | |
|------|------|------|------|------|
| | VE | MLE | VE | MLE |
| 32 | 987.4 | 446.5 | 780 | 600 |
| 64 | 664.7 | 560.9 | 800 | 620 |
| 128 | 994.8 | 959.8 | 710 | 800 |
| 256 | 418.8 | 994.1 | 700 | 890 |
| 500 | 980.87 | 628.66 | 600 | 830 |
| 1000 | 945.3 | 562.61 | 650 | 810 |

Table 10: size $\mathcal{V}$ (width) = 128

| rank | Our Results | | Reported Results | |
|------|------|------|------|------|
| | VE | MLE | VE | MLE |
| 2 | 152.9 | 112.3 | 180 | 60 |
| 4 | 524.2 | 224.7 | 780 | 100 |
| 6 | 981.0 | 726.6 | 800 | 400 |
| 8 | 994.3 | 927.1 | 790 | 690 |
| 10 | 994.8 | 959.8 | 800 | 740 |
| 12 | 880.7 | 981.9 | 690 | 700 |
| 14 | 684.5 | 972.3 | 700 | 760 |
| 16 | 842.7 | 974.2 | 690 | 780 |
| 20 | 671.1 | 993.8 | 800 | 800 |

These results are average returns for 100 consecutive episodes corresponding to the best policies formed using Double DQN.

### 4.2   Results beyond original paper

We used a different evaluation technique for the results of table 5. Instead of using approximate policy iteration with LSTD, we used Double DQN with linear action-value function approximators. The hyper-parameters used for this can be found in table 11.

Table 11:

| Hyper-parameters | Values |
|---|---|
| DQN learning rate | $10^{-3}$ |
| size of experience replay | $10^4$ |
| total iterations | $10^6$ |
| DQN learning frequency | 2 |

## 5  Discussion

Our results support the existence of Value Equivalent models in lower-dimensional spaces and the ability of Value Equivalence training to converge to these models.

Even though we were not able to reproduce the results for experiments with $\mathrm{span}(\mathcal{V}) \approx \tilde{\mathcal{V}}$ with linear function approximation, we have a few likely reasons behind this rather than the failure of VE models.

- The performance of both the VE and baselines models depend highly on the feature formation, which was done using k-means. The results of k-means are heavily dependent on the initialisation of centroids.
- The evaluation method not only diverged for the VE models but also for MLE models.
- Table 5 suggests that the problem could lie in the authors' evaluation technique, i.e. LSTD with approximate policy iteration.

### 5.1  What was easy

The paper's experiments were easy to follow and code, as the authors have divided the entire pipeline into different stages with clear descriptions for every stage.

### 5.2  What was difficult

Some points were left out or not evident in the original paper, and we had to ask the authors to get them cleared.

- Feature formation for experiments with $\mathrm{span}(\mathcal{V}) \approx \tilde{\mathcal{V}}$ and linear function approximation was done by applying k-means on a set of $10^6$ states collected using a random policy.
- The size of experience replay used in the paper was $10^6$.
- Rank constrained matrices where optimised by applying gradient descent to the parameters $F_D$ and $F_K$.

### 5.3  Future work

Grimm et al. [2020] uses rank constraints for Neural Networks in experiments with $\mathrm{span}(\mathcal{V}) \approx \tilde{\mathcal{V}}$. For experiments with ranks greater than 8, constraints on the first and last layer are effectively removed. This is because these layers each have one dimension, which is smaller than 8. It would be interesting to use other constraints like positive-semi definiteness (Lezcano-Casado [2019]) along with rank constraints to compare the performance of VE and MLE models.

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
