# OpenReview forum: "Reproducibility study of the Value Equivalence Principle for Model Based Reinforcement Learning"
_ML_Reproducibility_Challenge/2020 — Reject_

### Official Review · AnonReviewer2 · 2021-02-23
**valuable check of the original paper**

**Rating:** 6
**Confidence:** 4

**Review:**

The paper is a valuable check of the original paper.
A few corrections in formatting and language are needed.

span should not be italic, but rather {\rm span

A blank is missing in „fit(more“

„The number of experiments performed where realtively [sic] less“ -> „The number of experiments was relatively small“

„table[2]“ -> „Table 2“

„approximators.[table[5]]“ -> „approximators (see Table 5).“

I think, „5e-5“ should be replaced by 5\cdot 10^{-5}


**Familiar With The Original Paper:**

I have read the original paper

**Reproducibility Summary:**

Report has summary

---

### Official Review · AnonReviewer1 · 2021-03-02
**decent report**

**Rating:** 5
**Confidence:** 4

**Review:**

Reproducibility Summary : The report includes this summary as the first page, which contains major findings.
Scope of reproducibility: The report concisely and clearly states the scope, and follows it.
Code: As mentioned in the report that the original code was closed sourced, the authors reproduced the code for several sets of experiments.
Communication with original authors: The authors of this report contacted the original authors for multiple queries, and the original authors replied to these queries. However, I am not sure whether the original authors have evaluated the results in this report.
Hyperparameter Search: It seems that the authors of the report mostly used from the hyperparameters in the original paper.
Ablation Study: The ablation study is not comprehensive.
Discussion on results: The report discusses the state of reproducibility of the original paper, and mentions the easy parts and difficult parts. More specifically, for the case with linear function approximation, several results in the original paper could not be reproduced.
Recommendations for reproducibility: It seems the report does not discuss on how the original paper can improve its reproducibility.
Results beyond the paper: The report tries to show the results in Table [5] using a different evaluation technique, but it seems that the new results are not complete.
Overall organization and clarity: The organization is ok, but there are several grammatical issues, e.g., several periods are missing.

**Familiar With The Original Paper:**

I have read the original paper

**Reproducibility Summary:**

Report has summary

---

### Decision · Program_Chairs · 2021-03-31

**Decision:**

Reject

**Comment:**

Overall reviews and/or the paper content not good enough for the AC to recommend to the journal.